# Trans-Spinal Electrical Stimulation Therapy for Functional Rehabilitation after Spinal Cord Injury: Review

**DOI:** 10.3390/jcm11061550

**Published:** 2022-03-11

**Authors:** Md. Akhlasur Rahman, Niraj Singh Tharu, Sylvia M. Gustin, Yong-Ping Zheng, Monzurul Alam

**Affiliations:** 1Department of Biomedical Engineering, The Hong Kong Polytechnic University, Hong Kong, China; akhlas.physio@gmail.com (M.A.R.); singh.tharu@connect.polyu.hk (N.S.T.); yongping.zheng@polyu.edu.hk (Y.-P.Z.); 2Centre for the Rehabilitation of the Paralysed (CRP), Savar Union 1343, Bangladesh; 3NeuroRecovery Research Hub, School of Psychology, University of New South Wales, Sydney, NSW 2052, Australia; s.gustin@unsw.edu.au; 4Centre for Pain IMPACT, Neuroscience Research Australia, Sydney, NSW 2031, Australia

**Keywords:** spinal cord injury, rehabilitation, neuromodulation, transcutaneous electrical stimulation

## Abstract

Spinal cord injury (SCI) is one of the most debilitating injuries in the world. Complications after SCI, such as respiratory issues, bowel/bladder incontinency, pressure ulcers, autonomic dysreflexia, spasticity, pain, etc., lead to immense suffering, a remarkable reduction in life expectancy, and even premature death. Traditional rehabilitations for people with SCI are often insignificant or ineffective due to the severity and complexity of the injury. However, the recent development of noninvasive electrical neuromodulation treatments to the spinal cord have shed a ray of hope for these individuals to regain some of their lost functions, a reduction in secondary complications, and an improvement in their life quality. For this review, 250 articles were screened and about 150 were included to summarize the two most promising noninvasive spinal cord electrical stimulation methods of SCI rehabilitation treatment, namely, trans-spinal direct current stimulation (tsDCS) and trans-spinal pulsed current stimulation (tsPCS). Both treatments have demonstrated good success in not only improving the sensorimotor function, but also autonomic functions. Due to the noninvasive nature and lower costs of these treatments, in the coming years, we expect these treatments to be integrated into regular rehabilitation therapies worldwide.

## 1. Introduction

Spinal cord injury (SCI) is sudden, devastating, debilitating, and life-altering neurological damage that is correlated with severe physical, mental, social, and vocational impacts on the individual, family members, and healthcare systems [1,2,3,4,5]. Worldwide, the estimated incidence of SCI is between 250,000 and 500,000 per year [3]. Damage to the spinal cord can be permanent, which may result in either tetraplegia or paraplegia that may lead to functional impairment [5]. The American Spinal Injury Association (ASIA) Impairment Scale (AIS) classifies SCI into two categories: either complete or incomplete. AIS-A is defined as no sensory and motor function in the lowest sacral segment (S4–S5). In contrast, in incomplete (AIS-B, C, D, or E) SCI, there is some sensory and motor preservation as far as the lowest sacral segment (S4–S5) [6,7,8]. An individual with an SCI is at risk of different types of secondary complications which leads to premature death in most of the cases [9]. Secondary complications, such as pressure ulcers, spasticity, pain, urinary tract infections, depression, bowel problems, cardiovascular complications, renal stones, autonomic dysreflexia, fracture [10], fatigue, respiratory complications, autonomic dysreflexia, and bladder dysfunction, hamper the healthy wellbeing, social participation, and productive activities as well as quality of life [11]. Among all the complications, paralysis or reduced mobility are the most visible complications after SCI [1]. Reversing the paralysis often reduces the secondary complications and, thus, improves the quality of life of these individuals with SCI [12]. Currently, there is no standard neuromodulatory treatment for spinal cord injury (SCI). Conventional treatment approaches aim to reduce secondary complications and improve the functions that have been retained after the injury [13]. However, in recent years, various neuromodulatory approaches, such as epidural spinal cord stimulation (SCS) [14], intraspinal microstimulation, functional electrical stimulation (FES) [15] and transcutaneous spinal cord stimulation (tSCS) [16], have been studied extensively. Among them, noninvasive stimulations such as FES and tSCS are comparatively safer [17] and easier to implement in clinical settings than the invasive ones [18]. Noninvasive tSCS is a relatively new technology which targets the neural structure similar to its invasive counterpart, SCS [19]. However, depending on the stimulation parameters, tSCS can be broadly divided into two types: direct current stimulation (tsDCS) and pulse current stimulation (tsPCS). While research has been carried out on both of these stimulations by different research groups around the world, there is no study or review of objective comparisons between them. The goal of this study is to determine and compare the efficacy of these two noninvasive trans-spinal electrical stimulations for functional rehabilitation in SCI. This narrative review discusses tsDCS and tsPCS with a brief description of their usage, mechanisms, limitations, and implementation as well as subjectively compares them with other noninvasive stimulations, FESs, for SCI rehabilitation.

Topics related to SCI rehabilitation, neuromodulation, electrical stimulation in rehabilitation, new research in SCI rehabilitation, and publications in the English language were used to filter the 250 articles. Following the selection of articles, we extensively examined the abstracts and identified the most relevant articles based on the article’s aims and the criteria indicated previously. Finally, about 150 articles were selected based on SCI rehabilitation, which encompass motor recovery, electrical stimulation, and the mechanism, management, and treatments for SCI.

### 1.1. Conventional Treatments and Management after SCI

Current SCI rehabilitation depends on the injury and impairment level which concentrates on the most appropriate recovery from injury and helps a person to return to the community in a maximally independent way. This long, time-consuming rehabilitation process includes bed mobility practice, wheelchair propel training, strengthening programmes, range of motion exercises, passive movements, stretching exercises, bowel/bladder function management, and a higher level of functional activity [20,21].

Conventional rehabilitation is an undistinguishable, weak treatment protocol which has very few guidelines. Although rehabilitation scientists have covered some parts, there is still no universal guideline [12]. Generally, clinicians continuously concentrate on the emergence-based treatment for independence and planning of the discharge of patients as quickly as possible because of the limitation of inpatient support and funding. Conventional acute rehabilitation, however, may vary from area to area or even practitioner to practitioner [12,22].

Respiratory problems are one of the major secondary complications after cervical cord injury, where muscle becomes impaired which leads to a decreased vital capacity and coughing ability. Therefore, they turn into secondary complications such as pneumonia which is one of the major causes of premature death after cervical SCI. Conventional rehabilitation for pulmonary function improvement involves a combination of management such as positioning, breathing exercises, respiratory muscle strengthening, etc. [12]. From the physical and psychological perspectives, neurogenic bowel/bladder dysfunction is a major secondary complication for a person with SCI. The timely cleaning of catheters, regular intake of food containing fibre, adequate drinking of water, and the manual evacuation or use of suppository and oral medication are some regular routines in conventional bladder/bowel management [23]. Spasticity and pain are other common secondary complications after SCI. Severe spasticity may lead to decreased mobility, decreased activity of daily living, the development of contractures, and pressure ulcers. Passive stretching, weight-bearing exercises, positioning, and oral medications can minimize spasticity [24]. More than 80% of people suffer from pain after SCI resulting in emotional disturbance, decreasing community participation and finally the hampering of the quality of life. The active range of movement, passive stretching, hot compression, and oral medications are used to decrease the severity of pain [25].

Autonomic dysreflexia (AD)—a potentially life-threatening illness—is frequently caused by a spinal cord segment. Patients with SCI at the T6 level or higher may experience autonomic dysreflexia [26,27]. Resting blood pressure after SCI is frequently lower than usual (hypotension), while a sudden increase in blood pressure of as little as 20 mm Hg can cause life-threatening emergencies [28]. This hypertension, which is a symptom of a disorder known as autonomic dysreflexia (AD), affects 50–90% of SCI patients with tetraplegia or high paraplegia [26]. When it occurs, sitting in an upright position, taking off any restrictive clothes, socks and shoes, immediate catheterization, medication for decreasing blood pressure, etc., need to be performed in an urgent manner for AD management [26]. 

Fatigue is a complex condition that involves physical, behavioural, and psychological processes, and is a factor that is responsible for depression and self-resilience. It is also a common symptom for a person with SCI. Fatigue is also associated with decreasing positive feelings and responses to physical tiredness resulting in anxiety, decreased cognition, cardiac arrhythmia, and functional inactivity. Previous research works suggested that the reduction in poor social interactions can improve psychomotor and vocational functioning [29,30].

### 1.2. Functional Electrical Stimulation (FES) Therapy for SCI Rehabilitation

Functional electrical stimulation (FES) is a neurorehabilitation technology that applies short pulses of electrical stimulation during specific mimicking activities such as walking or cycling [15,31]. It can be applied in different areas of the body through the skin above the targeted muscle and peripheral nerve. It is more preferred to apply FES over the target muscle fibres rather than nerves as there is less chance of tissue damage and also achieves a higher success rate but with less electrical power [32]. FES pulses are primarily quantified according to three parameters, including pulse width, frequency, and amplitude. Square waveform patterns and 300 to 600-microsecond pulse width waveforms are generally applied to exert different effects on the targeted muscle. The frequency of the stimulation varies depending on the specific purpose. Normally, it ranges from 20 to 50 Hz, where a low frequency is usually applied to avoid muscle fatigue and to induce smooth muscle contraction with a tingling but comfortable feeling. The usual range of amplitude of FES is a few milliamperes to 100 mA, where a high amplitude has more potential to achieve a depolarizing effect leading to the activation of more muscle fibres. The range of amplitude also depends on the time and shape of the stimulation as well as the applied area of the body part [32]. There are numerous benefits of FES therapy in SCI rehabilitation, including the enhancement of residual muscle strength, increased flexibility and range of motion of the joints or limbs, as well as a reduction in spasticity, which leads to enhanced sensorimotor functions [32]. FES has a wide range of applications in rehabilitation, as it has both direct and indirect effects [30]. FES mostly enhances and brings back upper and lower limb muscle functions [31,32], improves cardiopulmonary fitness such as rising peak ventilation, thereby enhancing the airflow rate and airway pressure [33], and decreases neuropathic or nociceptive pain and spasticity which commonly arise after SCI [34]. Pressure ulcers are another common complication after SCI, where FES plays an important role to increase the pressure ulcer healing rate [35]. In addition, FES helps to control bowel/bladder dysfunction as well as improve erection and ejaculation dysfunction [36]. Furthermore, it was found that FES increases circulation and body metabolism, which ultimately results in an improved muscle mass and balance, and a controlled posture [36].

### 1.3. Spinal Cord Stimulation (SCS) Therapy for SCI Rehabilitation

Epidural spinal cord stimulation (SCS) is an effective neuromodulatory treatment which reduces chronic pain, reduces the severity of spasticity, increases specific rhythmic motor activity in the lower limbs, activates the respiratory muscles, improves bladder control, and enhances sensory nerve activity and the capacity to affect numerous organs either in the autonomic nervous system or viscero-somatic reflexes [14,37,38,39]. This invasive stimulation technique is a challenging innovation because it needs comprehensive education, training, and pre- and post-operative patient cooperation.

A few additional key points must be considered before SCS surgery, including if the patient needs any kind of antithrombotic therapy, suffers from coagulopathy, or has any kind of infection. Furthermore, all participants should be assessed regarding the cause of any psychiatric or psychological problem which can interrupt the effectiveness of an epidural SCS [40]. An epidural SCS can also experience some technical and clinical complications. The technical complications include the breakdown of devices, current leakage, and malfunction of rechargeable batteries, which can lead to failure [40]. In general, complications arise from an epidural SCS at a staggeringly high rate of about 28% to 42% [41]. The technical complication rate of the devices is around 5% [42]. Clinical complications include tissue damage, bleeding, and haematoma, plus infection, which is the most common (4% to 10%) [41,43,44]. 

To harness the benefits and to avoid the complications of an epidural SCS, a noninvasive spinal cord neuromodulation modality, called transcutaneous SCS, has been developed and investigated recently [45,46,47,48]. Transcutaneous stimulation can be induced alone or in combination with functional therapy, which leads to an increased motor function in patients with chronic paralysis [17,49]. It has the therapeutic potentiality to improve voluntary motor function, improve upper and lower motor limb muscle strength, improve standing posture, improve gait, decrease spasticity, and improve trunk strength and overall spinal function [50,51,52]. Usually, the electrodes are placed on the skin over the spinous processes of the vertebral column [18]. Transcutaneous SCS can modulate the spinal reflex of the lower limbs and activate the sensory–motor fibres of the anterior/posterior root [18]. Additionally, in the lower limbs, it can stimulate multiple muscles at the same time [46]. It is a safe therapeutic tool which is also appropriate for all SCI patients except those with skin irritation [17].

Based on the stimulation principles, transcutaneous SCS can be divided into two categories: direct current stimulation and pulse current stimulation. In the rest of this article, we summarized the methods, working mechanisms, and therapeutic effects of these two transcutaneous stimulation modalities in the functional rehabilitation of SCI patients.

## 2. Trans-Spinal Direct Current Stimulation (tsDCS)

Direct current (DC) stimulation is a noninvasive tool of neuromodulation which applies a simple, safe, effective, and painless constant stimulation [16,53,54,55,56]. DC stimulation can influence a change in the cortical and subcortical transmission of the spinal cord, resulting in a functional improvement after SCI. The impact of DC stimulation depends on the connected field related to the underlying neuron, neuronal junction, and polarization of electrodes such as anodal and cathodal stimulation [57].

Trans-spinal direct current stimulation (tsDCS) uses DC with a constant intensity during neuromodulation. tsDCS electrodes are generally thick (6 mm), rectangular (7 × 5), and have an alike longitudinal pair of saline-soaked synthetic sponges enclosed by electrolyte gel [16,54,55,56,58]. These electrical fields need to be applied directly over the spine [53,55,56,59]. The electrical current is sourced from a battery-driven steady current stimulator (Figure 1). Generally, two electrodes are positioned in different areas of the body: one is placed above the targeted area of the spinous process of the vertebra and the other is commonly applied over the posterior part of the limbs or shoulders. tsDCS consistently delivers a continual current of 1.5 mA to 2.5 mA for about 15–20 min [54,55,56,59]. The current density of 0.06 mA/cm^2^ and total charge density of 0.064 C/cm^2^ [16,59] are sufficiently low enough to avoid any potential tissue damage [58]. The current steps up to the maximal level within 10 s and in the same way decreases at the end of the treatment session [54]. It induces changes in spinal cord physiology, nociceptive pathways, and cortical excitability, as well in the supraspinal centres [53,54]. On the other hand, large dissimilarities can be observed according to the participant’s physiology and anatomical parameters, such as age, sex, height, and weight [60,61]. The use of tsDCS is limited due to the large range of electrical currents that can be used in clinical settings [16] and the location of the electrodes, as even a 1 cm displacement can have a significant impact on the projected current flow pattern [62]. Headaches, weariness, vasodilation generating erythema, tingling, and itchiness are frequently reported side-effects [62].

### 2.1. Mechanism of tsDCS

tsDCS is one of the neuromodulatory tools which work directly on the neurons and have long-term effects on the spinal cord [57]. The exact mechanism of tsDCS is not fully understood, but a few researchers found that tsDCS on resting membranes acts equally for long-term potentiation and long-term depression. Moreover, tsDCS also causes a change to glutamatergic neurotransmission (a chemical that nerve cells use to send signals to other cells) [63]. Glutamate facilitates the pathway of phrenic motor neurons which provide complete motor innervations of the respiratory diaphragm, and consist of motor, sensory, and sympathetic nerve fibres, as well as N-methyl-D-aspartate (NMDA) receptors, which are very important for controlling synaptic plasticity and memory function [64]. Therefore, it leads to the neuronal activation process of the spinal cord. As a result, long-haul potentiation occurs in postsynaptic spinal cord motoneurons [58]. tsDCS stimulates neurotransmitters in the brain or in the spinal cord and also stimulates neural function through the ascending spinal pathways. Conductive elements of the corticospinal tract change after tsDCS; therefore, the resting motor threshold also changes [55]. tsDCS interacts between two functional neural circuits [63], which modulates the inhibitory tone and could be a therapeutic tool to decrease spasticity [55]. tsDCS may also activate the supraspinal loops which are transmitted by the brainstem or thalamo-cortical frameworks, followed by both ascending and descending tract inhibition [65]. 

Inactive motor neurons are activated after the use of tsDCS, which improves the connectivity between two functional neural circuits and a remarkable change comes about through the spinal motor neurons [66]. This helps to promote the inhibitory tone which is responsible for decreasing the severity of spasticity resulting from SCI. Additionally, tsDCS depends on the polarity of the stimulation, in relation to which researchers have found that cathodal stimulation activates neurons, whereas anodal stimulation depresses neurons [67]. With the use of anodal tsDCS stimulation, the nociceptive pathway decreases its conductivity and, subsequently, balances somatosensory-evoked possibilities which results in a decrease in pain sensations [68,69].

### 2.2. tsDCS in SCI Rehabilitation

In SCI rehabilitation, tsDCS has a significant implication on the promotion of axonal regeneration and is also responsible for preventing fibre degeneration [70]. Chronic SCI causes automatic micturition and neurogenic detrusor overactivity in both animals and humans. Following SCI, synchronized activity is lost, resulting in detrusor sphincter dyssynergia and inadequate bladder discharge, urological complications such as urinary tract infections, and chronic renal failure as a result of this process [71,72]. A cathode tsDCS has a significantly altered effect on sacral spinal reflexes and spinal neuronal networks, as well as bladder and external urinary sphincter functions in mice with an intact spinal cord and SCI [73,74]. Furthermore, tsDCS is an important neuromodulation therapeutic intervention which has been found to be safe, and no harmful effects on spinal-specific or other parts of the body have been observed after stimulation [70]. 

More than 80% of SCI patients have reported that they had been suffering from pain and 70% reported their suffering from spasticity after injury which had hampered their ability to partake in activities of daily living [11,75,76]. tsDCS is a simple and low-cost neuromodulation technique used in rehabilitation after neurological injury [16]. The Hoffman reflex (H-reflex) is usually utilized as a test to examine spinal inhibitory intraneuronal circuits. There is a relation between the changing of the sensitivity of the H-reflex and improving new motor function as well as new motor skills [54]. Albuquerque et al. and Awosika et al. [53,54] used tsDCS combined with motor training to decrease plasticity and maximize motor recovery for patients with neurological problems. Bocci et al. and Hubli et al. [56,77] worked with tsDCS by combining it with locomotion, where they found an improvement in gait after the use of tsDCS. The rhythmic activity of the brainstem central pattern generator leads to an increase in respiratory muscle contraction and duration; although tsDCS cannot increase the inspiratory time, it can increase the tidal volume [58]. tsDCS can facilitate cortical sensitivity and has been suggested as a promising solution for the comprehensive extension of neurological and psychiatric disorders [63]. It also helps to decrease pain severity, increase pain tolerance [55,56], reduce spasticity, and promote functional movements [60,61,64] (Kuck) [66]. Findings from various studies suggest that tsDCS is a noninvasive neuromodulation technology that can prevent neuronal dysfunctions which may develop after SCI. A recent study [77] reported that after a session of anodal tsDCS and assisted locomotion for a person with motor complete SCI, the spinal reflex amplitude improved, while the spinal reflex of upper limit decreased. This study also suggested that tsDCS might influence somatosensory, nociceptive, as well as motor pathways in a person with SCI. The thalamus, anterior cingulate, insula, and primary somatosensory cortices become active during painful stimulation, where tsDCS has an altered effect on such multicircuit junctions in the brain. tsDCS may diminish pain, decrease mechanical pain, improve temporal summation limits, and improve cold pain resistance [78,79,80,81]. The spinal cord is the centre through which ascending and descending neural signals pass. tsDCS causes the cortical area to depress and the corticospinal area to excite, so that short- and long-term effects are perceived in the cortico-phrenic and in the lemniscal pathways. Mainly, the soleus H-reflex becomes excited by cathodal tsDCS and depressed by anodal tsDCS, resulting in homosynaptic depression. Moreover, presynaptic resistance and postactivation depression can be stimulated by tsDCS and can decrease spasticity by modulating interneuronal sensitivity [73,74,82,83]. In another study, Niérat et al. [63] found that the cathodal part of tsDCS increases the respiratory neuromechanical element resulting in an increased tidal volume.

## 3. Trans-Spinal Pulsed Current Stimulation (tsPCS)

In trans-spinal pulsed current stimulation (tsPCS), an electric current is applied through electrodes superficially placed on the skin surface over the spine (Figure 2), targeting the spinal cord [82] in order to produce therapeutic effects [84]. A pulsed current is described as a one- or two-way movement of charged elements, where the current intensity varies frequently over time [16]. tsPCS is a noninvasive procedure [85] which modifies the action of neural connections on spinal stimulation [86], which is also an emerging method used for the study of neurologically impaired individuals [87]. In recent decades, tsPCS has mainly focused on pain management, whereas it is currently used as an important aspect of rehabilitation, commonly used for motor function restoration [88] in individuals with neural impairments [89] caused by SCI [90]. This stimulatory treatment is used commonly for damaged spinal cords for modulating the neural linkages [91] along with healthy individuals for various spinal-related studies [92]. tsPCS is easier to use and convenient for clinical purposes [18], where a range of currents at a frequency (5–50 Hz) and different intensities (10–200 mA) embedded in a carrier frequency of 5 and 10 kHz produce therapeutic benefits [93]. Regarding the efficiency and safety of stimulation currents, the pulsed current revealed better results than the direct current when applied to observe the neuropathologic changes after nerve injury [94]. Additionally, tsPCS with a larger pulse interval produces better outcomes than a shorter interval, and a biphasic pulse decreases the potential hazards by reducing the charge imbalance and skin injury [95]. Furthermore, the use of multisite stimulation results in more efficiency than one site of stimulation in the neuromodulation of the neural spinal connections in SCI [96], while single-site stimulation at the C5, T11, and L1 levels produces a higher magnitude than double-site stimulation when the lower extremities are in a gravity-neutral position [97]. tsPCS triggers a large to medium diameter of afferent fibres within the lumbar and upper sacral posterior roots [98], where a shorter burst period (100 µsec) produced a better response during stimulation than with a longer burst period [99]. Additionally, a stimulation intensity of higher than 20 mA was reported to increase the temperature of adjacent tissues, while larger electrodes produce a smaller rise in temperature [100]. Stimulation at a higher intensity produces some pain and discomfort which could be reduced using a particular waveform of 0.1–1 ms bursts of biphasic rectangular pulses with a carrier frequency of up to 10 kHz (Figure 2); however, the pain increases with stimulation above 20 kHz [101]. An intensity of more than 100 mA is delivered by a 10 kHz carrier frequency and is suitable for stimulating the spinal circuits of both injured and noninjured individuals [102]. The limitation of tsPCS depends upon the anatomical space over the stimulation region as it may induce changes in the delivery [103] and period of stimulation due to the chance of overheating the skin [104]. Some studies have reported a rise in systolic blood pressure of over 60 mmHg with skin rupture in the stimulation region [105]. 

### 3.1. Mechanism of tsPCS

The functional mechanisms of tsPCS have been studied in recent years. The transcutaneous electrodes of tsPCS produce an electric field which triggers the neural connectome [106] and the continuous stimulus increases the excitatory transmission in dorsal horn neurons with the inhibition of neuronal network dynamics that help to release endogenous neurotrophic factors which assist motor responses [107]. tsPCS targets the proximal afferent fibres of posterior roots to enhance rhythmic motor outputs [108]. The spinal cord is a junction of various spinal tracts and stimulation at the thoracic level has the potential of modulating the tracts [82]. The stimulation decreases the somatosensory-evoked potential responses in both the primary and secondary somatosensory cortices [109]. tsPCS induces motor action potentials in the upper and lower extremities which are prone to spinal inhibitory processes at the pre- and post-alpha motoneuronal level [110]. It uses the neural paths that carry descending drive and sensory inputs from the periphery to the spinal neuronal circuits to produce motor responses [107]. During tsPCS stimulation, the electrode length (5 × 5 cm) delivers a maximum output as it produces a lesser current intensity between the electrode and placement site [111], and single-pulse stimulation engages the higher to middle diameter of dermal and muscle sense afferent dorsal roots of various spinal connections, and is reliant on the stimulus intensity, since a 5 kHz carrier frequency promotes more physical improvement in the upper extremities [93]. 

When the damaged spinal connection responds to the stimulus, it indicates the potential for conversion into a functional state, which is sufficient for the restoration and improvement of motor functions [112]. The lower stimulation intensity results in the preferential selection of lower threshold afferent fibres that accompany some motor axons; thus, the rising intensity triggers additional motor axons, thereby causing a reduced latency of the response due to the occlusion effect of afferent paths [97]. The tsPCS promotes neuromuscular improvement in SCI after neural deficits [113] by modulating spinal neurons to regain motor improvement [86]. A few studies have claimed that in cervical SCI, sensorimotor spinal tracts could be modulated through stimulation [113], thereby enabling improvements to be achieved in some cognitive aspects [114], with restorative improvement caused due to the linkage of the propriospinal neurons of cervical networks [115]. Similarly, for injuries at the thoracic level, stimulation has been shown to cause persistent changes in trans-synaptic neurons that can assist in the prevention of neural impairments [16]. However, the stimulation of spinal connections at the lumbar level modifies the posterior roots and may vary according to current intensity, with changes in sites due to the functional and mechanical characteristics of the vertebral column [116]. The tsPCS develops communication amongst the motor neurons through spinal excitability produced by the activation of sensory afferents [117], and a frequency of 0.2 Hz caused changes in the lateral motor neurons once the stimulation electrodes were positioned on the same side medial to the spine [118]. Interestingly, the sensory roots are stimulated during tsPCS stimulation in supine and standing positions, whereas motor roots are triggered in the prone position [119]. A higher carrier frequency was also found to be effective in developing muscle strength [93]. The double stimulus used in a closer sequence revealed the depression of the effects of the second stimulus across all muscles [40]. Simultaneously, a stimulus with a lower intensity included small threshold afferents, which assist the weak selection of motor fibres, whereas a rise in the stimulation intensity resulted in an increased motor fibre activation [120]. 

### 3.2. tsPCS in SCI Rehabilitation

In contrast to classical rehabilitation interventions, recent applications of tsPCS have opened up a new horizon of functional recovery from paralysis and other neurological conditions after SCI [16]. tsPCS at greater intensities (>80 mA) activates the trunk musculature which helps to maintain posture [18] and the duration of stimulation should be no less than 20 min to produce therapeutic effects [117]. Simultaneously, tsPCS at the L1–L2 sites at 15 Hz enables the extension of the trunk muscles leading to postural stability [19], while at 30–50 Hz, stimulation for more than 20 min evokes responses and produces a greater possibility of functional improvement for trunk control in SCI [47]. During tsPCS stimulation, the movement of flexor muscles exhibits an increment during the delivery of a higher frequency (80–100 Hz), whereas a lower frequency (20–30 Hz) increases the extensor group movement [18]. Locomotor training revealed controlled flexion during the modulation of the upper lumbar cord, while a higher sacral stimulation exhibited extension during stepping [121]. Another study showed that the stepping action is initiated by tsPCS with an intensity of 50–80 mA for the production of locomotor activity in SCI [122] and numerous motor pools can also be modulated and coordinated by imagined stepping [123]. Similarly, controlled function such as leg extension was elicited when the rostral part of T11–T12 was stimulated at 30 Hz and L1–L2 of the caudal zone at 15 Hz [19]. Another study revealed an improvement of locomotion activity when applying tsPCS on rostral sites, while stimulating the caudal area aided restorative extensor movement of the trunk muscles [124]. Similarly, when the position of electrodes was changed to the innervation region of the biceps brachii (C5–C6) muscle, it led to the activation of joint movement, thereby improving the elbow’s range of motion [125]. The stimulus to the lateral side of T11 produced flexion, while at L1 it led to extension during walking [126], and the stimulation of upper limbs exhibited prolonged changes in neural linkages with continued recovery of hand and arm function in SCI [85]. Spinal stimulation at multiple sites was found to be effective during functional activities such as locomotion and standing, while region-specific stimulation modulated the upright standing posture [127]. tsPCS at lower intensities (5.5–51 mA) caused the activation of the proximal arm and shoulder muscles [95], showing immediate improvements in handgrip strength [128] and upper limb motor control in individuals with tetraplegia [84]. 

tsPCS at 30 Hz enhanced motor movement of the ankle in SCI in long-term injuries [51] and at 50 Hz exhibited a reduction in spasticity of the lower limbs in chronic SCI [129], whereas at 25–60 Hz, it aided lower extremity movement [40,130]. It also demonstrated locomotor development upon the stimulation of the sensory and motor roots of the lower extremities at a 30 Hz intensity, whereas 50 Hz produced paraesthesia among people with sensory incomplete SCI [131]. The use of tsPCS for the bladder with stimulation at 15 Hz decreased the voiding volume by 33–26 mL, whereas it increased when stimulated at 10 Hz [132]. It also reduced detrusor overactivity, increased bladder capacity, and reduced episodes of incontinence [133], while stimulation at 50 Hz at the thoracic level (T9–T11) resulted in improvements in the bowel management routine [134]. The placement of reference electrodes on the abdominal area with active electrodes over the thoracic and lumbar spinal networks during tsPCS produced the activation of pelvic and urinary control muscles [135]. It also improved respiratory functions such as the breathing and coughing ability, with increased functionality of the trunk muscles [136]. tsPCS triggers sensory nerve roots through antidromic stimulation and assists vasodilation, leading to a rise in blood circulation of the skin [137], and stimulation at the thoraco-lumbar level (T12–L1) instead of the cervical (C5–C6) at 30 Hz causes a rise in microcirculation [138], and a decreased response in the muscle spindles of afferent fibres was observed after stimulation of the long tracts of the spinal cord in individuals with SCI [139]. 

## 4. Conclusions

Neuromodulation is rapidly growing in therapeutics and affecting thousands of patients with various disabilities around the world. Electrical neuromodulation is a harmless and well-studied technology that can be easily programmed for the excitation, inhibition, and modification of the functioning of the nervous system. Noninvasive spinal cord neuromodulation, namely, tsDCS and tsPCS, uses transcutaneous electrical stimulation through the spinal cord and its peripheral nerves, and is an effective treatment protocol for SCI and other neurological conditions, including stroke, chronic pain, spasticity, respiratory disorders, cardiovascular ischaemia, neuropathic bladder, bowel dysfunction, and upper and lower limb function, including fine motor function with digit function. Increasing evidence has demonstrated that tsDCS and tsPCS could be potentially effective treatment options for patients suffering from SCI. However, the major complexity in such a neuromodulatory treatment is the evaluation and comparison of clinical outcomes in both short and medium/long terms. While short-term effects are quite promising, the medium- or long-term effects of such a neuromodulation have not yet been evaluated and are very difficult to predict. They also vary from person to person as no injury or condition is exactly the same even though falling under the same cohort. Moreover, evidence is growing that the improvement in sensorimotor functioning in spinal cord injuries is associated with sensorimotor cortical plasticity [140,141,142]. Hence, sensorimotor cortical activation measured by functional magnetic resonance imaging (fMRI) may be a potential biomarker for the restoration of sensorimotor function [141,142]. Indeed, there is evidence that somatosensory cortical activation measured by fMRI may become a biomarker for the detection of preserved somatosensory pathways, and, hence, may also be used to objectively measure sensorimotor improvement [143]. More studies on the effects of the specific stimulation parameters of tsDCS and tsPCS on the functional rehabilitation of SCI would provide a better clinical guideline. Furthermore, a systematic review and meta-analysis of homogeneous patient cohorts (e.g., similar severity, type, and injury of SCI patients) would be beneficial to understand the effectiveness and adverse effects of tsDCS and tsPCS.

## Figures and Tables

**Figure 1 jcm-11-01550-f001:**
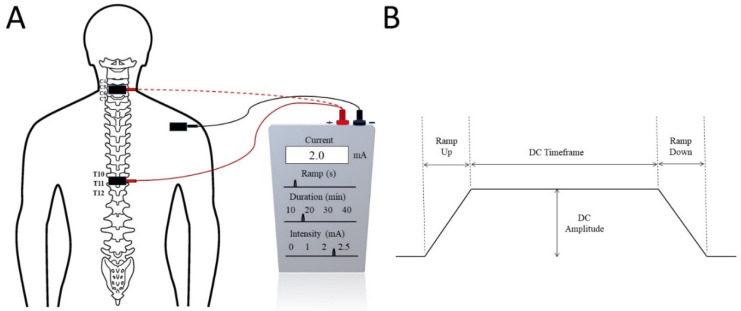
(**A**) Trans-spinal direct current stimulation (tsDCS) at the cervical (dotted red) or thoracic (solid red) spinal level with the reference electrode commonly placed on the shoulder. (**B**) Typical tsDCS signal with several seconds of ramp up and ramp down current. The treatment intensity and duration are adjusted by the DC Amplitude and DC Timeframe.

**Figure 2 jcm-11-01550-f002:**
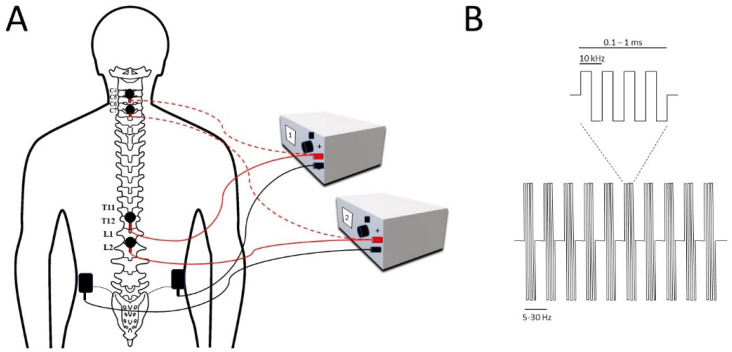
(**A**) Multisite trans-spinal pulsed current stimulation at the cervical (dotted red) and thoracolumbar (solid red) levels with reference electrodes commonly placed above the iliac crests. (**B**) Typical tsPCS signal delivered at a frequency of 5–30 Hz with a 0.1–1 ms pulse width modulated at 10 kHz biphasic stimulation.

## Data Availability

Not applicable.

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
