# Peer review of "Trans-Spinal Electrical Stimulation Therapy for Functional Rehabilitation after Spinal Cord Injury: Review"

_jcm, 2022, doi:10.3390/jcm11061550_

Round 1
Reviewer 1 Report
The authors present a review about trans-spinal electrical stimulation therapy for functional rehabilitation after spinal cord injury. This study is interesting and brings knowledge in the area. However several issues should be clarified by the authors.
In any part of the manuscript the authors presented the aim of the manuscript. Also, the introduction session also fails in demonstrating the investigation problem that sustain the purpose of the manuscript. To clarify the readers, the authors should also include a paraghraph explaining the organization of the manuscript.
From a methodological point of view it is not clear for me why the authors didn´t performed a systematic review according to PRISMA recommendations. The authors make conclusions about effectiveness of trans-spinal eletrical stimulation therapy in functional rehabilitation after spinal cord injury like “summarise the two most promising non-invasive spinal cord electrical… Both treatments have shown good success on not only improving the sensorimotor function but also autonomic functions”. However, the study design doesn´t support these findings. What was the criteria for the screening of the 250 articles and the criteria for selecting 150 articles for the review?
The non clarity of the aim of the study turn dificult understanding the relation between the different sessions of the manuscript and its aim.
The english writing should also be revised for gramatical acuracy and clarity.
Without these clarifications it is dificult for me to give a more specific feedback.
Author Response
The authors present a review about trans-spinal electrical stimulation therapy for functional rehabilitation after spinal cord injury. This study is interesting and brings knowledge in the area. However several issues should be clarified by the authors.
In any part of the manuscript the authors presented the aim of the manuscript. Also, the introduction session also fails in demonstrating the investigation problem that sustain the purpose of the manuscript. To clarify the readers, the authors should also include a paraghraph explaining the organization of the manuscript.
Response: Thanks for the kind evaluation of the manuscript and your valuable comments. To facilitate the readers on the organization of the manuscript, we have added the following paragraph in the introduction section:
Currently, there is no standard neuromodulatory treatment for spinal cord injury (SCI). Conventional treatment approaches aim to reduce secondary complications and improve the functions that have been retained after the injury [1]. However, in recent years, various neuromodulatory approaches, such as epidural spinal cord stimulation (eSCS) [2], intraspinal microstimulation, functional electrical stimulation (FES) [3] and transcutaneous spinal cord stimulation (tSCS) [4] have been studied extensively. Among them the non-invasive stimulations such as FES and tSCS are comparatively safer [5] and easier to implement in clinical settings than the invasive ones [6]. Non-invasive tSCS is a relatively new technology which targets the neural structure similar to it’s invasive counterpart, eSCS [7]. However, depending on the stimulation parameters tSCS can be broadly divided into two types: direct current stimulation (tsDCS) and pulse current stimulation (tsPCS). While research has been carried out on both the stimulations by different research groups around the world, there is no study or review of objective comparisons between them. The goal of this study is to determine and compare the efficacy of these two non-invasive trans-spinal electrical stimulations for functional rehabilitation in SCI. This narrative review discusses tsDCS and tsPCS with a brief description of usage, mechanisms, limitations, and implementation as well as subjectively compares with other non-invasive stimulation, FES for SCI rehabilitation.
From a methodological point of view, it is not clear for me why the authors didn´t performed a systematic review according to PRISMA recommendations.
Response: The focus of the review is on non-invasive emerging techniques which are being developed as a replacement for the invasive neuromodulation methods. They are also considered to be promising and important in the rehabilitation of spinal cord injuries. However, the research findings are still in the very early stage. Furthermore, studies that utilised these techniques used a largely variable pull of cohorts (different severity and injury of SCI) which cannot be systemically compared. Thus, we believe that there is too much variation in the limited number of small-scale studies to feel particularly confident in the value of a systematic review.
In comparison to the conventional treatment, we think that the techniques presented in this study could be a viable treatment alternative, hence would like to introduce the aspects of these treatment modalities to a broad audience, whereas a systematic review is designed to evaluate and compare the effects of treatments on same cohorts. Therefore, we did not perform a systematic review instead presented a narrative review. However, we also agree with the reviewer that in the future a systematic review on more specific patient cohorts would benefit the readers. We have included the following sentence in the Conclusion section to reflect this suggestion: Furthermore, a systematic review and meta-analysis on homogeneous patient cohorts (e.g., similar severity, type and injury of SCI people) would be beneficial to understand the effectiveness and adverse effects of tsDCS and tsPCS.
The authors make conclusions about effectiveness of trans-spinal eletrical stimulation therapy in functional rehabilitation after spinal cord injury like “summarise the two most promising non-invasive spinal cord electrical… Both treatments have shown good success on not only improving the sensorimotor function but also autonomic functions”. However, the study design doesn´t support these findings. What was the criteria for the screening of the 250 articles and the criteria for selecting 150 articles for the review?
Response: To facilitate the readers on the study design, we have added following paragraph in the revised manuscript:
Topics related to SCI rehabilitation, neuromodulation, electrical stimulation in rehabilitation, new research in SCI rehabilitation, publications in the English language were used to filter the 250 articles. Following the selection of articles, we extensively examined the abstracts and identified the most relevant articles based on the article's aims and the criteria indicated previously. Finally, about 150 articles were selected based on SCI rehabilitation, which encompasses motor recovery, electrical stimulation, mechanism, management, and treatments for SCI.
The non clarity of the aim of the study turn dificult understanding the relation between the different sessions of the manuscript and its aim.
Response: The present study analyses and contrasts previously used stimulation approaches in spinal cord injury rehabilitation, subjectively demonstrating how the current emerging spinal cord stimulation (SCS) techniques outperform the old ones. Because transcutaneous SCS is relatively a new technology, we recognize the necessity of discussing the process and potential mechanisms, hence have gone indepth regarding the two stimulation procedures. Furthermore, the effect of such strategies has been found to be helpful in increasing sensorimotor function and, even in autonomic functions, as summarised in the manuscript.
The english writing should also be revised for gramatical acuracy and clarity.
Response: The revised manuscript is proofread and corrected by a professional English editor.

Reviewer 2 Report
The Authors provide a comprehensive review on trans-spinal direct current stimulation and trans-spinal pulsed current stimulation, two promising tools for non-invasive neuromodulation and rehabilitation after spinal cord injury. The revision is appropriate for the Special Issue "Spinal Cord Injuries: Functional Challenges and Latest Therapeutic Advances", for which it was probably conceived.
I only have few minor issues to be addressed. Limitations of the presented techniques, both in methodological application and results interpretation standardization, have not been fully elucidated; this point should be addressed. I would suggest to stress that the major complexity in such neuromodulatory therapy is to evaluate and compare clinical outcome, in both the short and medium/long term; which are the most conditioning factors? Also investigate the possible role for imaging techniques (functional MRI? tractography? nuclear medicine?) in assessing morpho-functional variations related to trans-spinal electrical stimulation (is there any potential biomarker? which are future perspectives in monitoring DSC/PSC?); please expand. Moreover, consider to propose a meta-analysis of the tabulated studies (DSC vs PSC), as it would probably be more interesting for Clinical Neurology readers to have a comparison between these two techniques.
Finally I would suggest to move the tables in supplementary materials, as they are difficult to read in their present graphic form.
Author Response
I only have few minor issues to be addressed. Limitations of the presented techniques, both in methodological application and results interpretation standardization, have not been fully elucidated; this point should be addressed.
Response: Thanks for the kind evaluation of the manuscript and your valuable comments. We have added the following paragraphs to adhere on explaining the limitations of the presented techniques:
The methodological limitation of tsPCS depends upon the anatomical space over the stimulation region as it may induce changes in the delivery [8] and period of stimulation, due to chances of overheating the skin [9]. Some studies have reported of rising in systolic blood pressure over 60 mmHg with a skin rupture in the stimulation region [10].
The use of tsDCS is also limited due to the large range of electrical currents that can be used in clinical settings [4] and the location of the electrode, as even, a 1 cm displacement had a significant impact on the projected current flow pattern [11]. Headache, weariness, vasodilation generating erythema, tingling, itchy sensations, and other side effects are frequently reported [11].
I would suggest to stress that the major complexity in such neuromodulatory therapy is to evaluate and compare clinical outcome, in both the short and medium/long term; which are the most conditioning factors?
Response: Thank you so much for your valuable suggestion. We have added the following statement in the revised manuscript:
However, the major complexity in such neuromodulatory treatment is to evaluate and compare the clinical outcome, in both short and medium/long term. While short term effects are quite promising, the medium- or long-term effects of such neuromodulation have not yet been evaluated and are very difficult to predict. It also varies for person to person as no injury or condition are exactly the same even though they may fall under the same cohort.
Also investigate the possible role for imaging techniques (functional MRI? Tractography? Nuclear medicine?) in assessing morpho-functional variations related to trans-spinal electrical stimulation (is there any potential biomarker? Which are future perspectives in monitoring DSC/PSC?); please expand.
Response: Thanks for this valuable comment. We have included the following paragraph in the revised manuscript:
Moreover, evidence is arising that improvement in sensorimotor function in spinal cord injury is associated with sensorimotor cortical plasticity [12-14]. Hence, sensorimotor cortical activation measured by functional magnetic resonance imaging (fMRI) may represent a potential biomarker for restoration of sensorimotor function [13, 14]. Indeed, there is evidence that somatosensory cortical activation measured by fMRI may present a biomarker for the detection of preserved somatosensory pathways, and hence may be also used to objectively measure sensorimotor improvement [15].
Moreover, consider to propose a meta-analysis of the tabulated studies (DSC vs PSC), as it would probably be more interesting for Clinical Neurology readers to have a comparison between these two techniques.
Response: Thank you for this comment. We believe that there is too much variation in the limited number of small-scale studies to feel particularly confident in the value of a meta-analysis. The heterogeneity in terms of the breadth of different cohorts (e.g., different severity and injury of SCI people) makes it difficult lumping these all together. However, we agree with the reviewer that in the future a meta-analysis on more specific patient cohorts would be beneficial. We have included the following sentence in the Conclusion section to reflect this suggestion: Furthermore, a systematic review and meta-analysis on homogeneous patient cohorts (e.g., similar severity, type and injury of SCI people) would be beneficial to understand the effectiveness and adverse effects of tsDCS and tsPCS.
Finally I would suggest to move the tables in supplementary materials, as they are difficult to read in their present graphic form.
Response: Thanks for the kind suggestion. We have moved the tables to the supplementary file in the revised manuscript.

Reviewer 3 Report
The authors described the concise review about trans-spinal electrical stimulation for spinal cord-injured patients. The present manuscript is comprehensive and well covers this field. The reviewer feels satisfied to read this manuscript.
Author Response
We sincerely thank the reviewer for the kind evaluation of our manuscript. To improve the grammar and spell check, the revised manuscript is proofread and corrected by a professional English editor.

Reviewer 4 Report
It is evaluated as a well-written review article on transspinal electrical stimulation for functional rehabilitation in patients with spinal cord injuries.
Author Response

(The authors gave the same response as above.)
